# Research on the mechanical properties of hybrid Steel Fiber-Reinforced Self-Compacting Concrete based on the DIC method

Hang Hu[1], Jianyi Gu[2]*, Yang Tu[3], Yi Zhou[3], Fuwei Zhu[3]

1 Chengdu Communications Investment Group Co., Ltd, Chengdu, China, 2 School of Civil Engineering, Chongqing Jiaotong University, Chongqing, China, 3 Chengdu Road & Bridge Operation and Management Co., Ltd. Cheng Du, China

* gujianyi@163.com

## Abstract

To investigate the influence of hybrid steel fibers on the mechanical properties of self-compacting concrete, this study systematically compared the mechanical performance of ordinary self-compacting concrete, single steel fiber-reinforced self-compacting concrete, and three types of hybrid steel fiber-reinforced self-compacting concrete through cube compression tests, four-point bending tests, and elastic modulus tests. The digital image correlation (DIC) method was employed to accurately measure the surface displacement and strain of the specimens. The results indicate that the self-compacting hybrid steel fiber concrete specimens exhibited excellent flexural-tensile performance. The incorporation of steel fibers significantly inhibited the development of macroscopic cracks, enhanced the crack resistance of the concrete, and effectively improved its brittleness. Particularly, when the fiber content was 1% 6mm + 1% 13mm + 1% 25mm, the compressive strength increased by 41.15%, the flexural strength increased by 266.41%, and the elastic modulus reached 40.82 MPa. Using the DIC method, this study successfully captured the full-field strain evolution during the failure process of the specimens. Based on the evolution of the maximum principal strain, the dynamic evolution characteristics of macroscopic and microscopic cracks on the specimen surface were thoroughly analyzed. The study found that the enhancement effect of hybrid steel fibers (HF) on concrete is primarily manifested in hindering the propagation of macroscopic cracks. This research provides important theoretical foundations and practical guidance for the engineering application of self-compacting concrete.

## 0 . Introduction

**Self-Compacting Concrete (SCC)** has been widely used in modern civil engineering due to its high fluidity, self-compacting properties, and excellent construction

**Data availability statement:** All relevant data are within the manuscript and its Supporting Information files.

**Funding:** The author(s) received no specific funding for this work.

**Competing interests:** The authors have declared that no competing interests exist.

performance [1]. However, the inherent brittleness and low tensile strength of SCC limit its application in high-performance engineering projects requiring crack resistance and seismic performance [2]. To address these limitations, the incorporation of fibers has emerged as an effective reinforcement method [3–6].

In recent years, numerous studies have demonstrated the significant advantages of **Steel Fiber-Reinforced Self-Compacting Concrete (SFRSCC)** in terms of mechanical properties[7–10]. The addition of steel fibers can notably enhance the compressive strength, flexural strength, and fracture toughness of concrete, while also improving its fatigue resistance and impact resistance. Gao Danying et al. [11] investigated the effects of different steel fiber contents and types on concrete strength, revealing that steel fibers moderately improve the compressive strength of SCC. Liu Siguo et al. [12] analyzed the influence of steel fibers on the compressive strength of SCC with different strength grades, They established a predictive model for 28-day compressive strength based on experimental data. Cai Canliu et al. [13] studied the effects of varying lengths of corrugated steel fibers on the compressive and flexural strength of SCC, finding that longer steel fibers significantly enhance these properties, particularly at a fiber volume fraction of 1%. Long Wujian et al. [14] examined the impact of various steel fiber types on the compressive and splitting tensile strength of SCC with different strength grades (C30, C40, C50). Their results indicated that while the strength of steel fiber-reinforced SCC was lower than that of plain concrete at 7 days, it exceeded plain concrete at 28 days, with hooked-end fibers providing the greatest improvement in splitting tensile strength. Majain et al. [15] analyzed the effects of steel fiber volume fractions (0%, 0.5%, 1.0%) on the compressive strength of SCC, showing that the compressive strength increased by 6.6% and 8.0% at 0.5% and 1.0% fiber content, respectively. Siddique et al. [16] found that steel fibers significantly enhance the mechanical properties of SCC, particularly splitting tensile and flexural strength, more so than compressive strength. Khaloo et al. [17] observed that while steel fibers may slightly reduce compressive strength, they effectively improve splitting tensile and flexural strength.

However, most current research focuses on the mechanical performance enhancement of SCC using two types of steel fibers with different aspect ratios. For example, Akcay et al. [18] studied the effects of steel fibers with different aspect ratios by incorporating 1% straight steel fibers (6 mm length, 0.15 mm diameter) and 0.5% hooked-end steel fibers (30 mm length, 0.55 mm diameter) into SCC. Their results showed that these fibers dispersed uniformly without clumping, significantly improving the toughness of the concrete. Deeb et al. [19] investigated the workability of SCC with high-volume hybrid steel fibers (5% short fibers: 6 mm length, 0.16 mm diameter; 1% medium-length fibers: 13 mm length, 0.16 mm diameter), demonstrating that such mixtures can achieve the required fluidity for SCC, although the study did not explore the mechanical performance enhancement. Dimas et al. [20–22] used 0.5% straight steel fibers (12 mm length, 0.18 mm diameter) and 1.5% hooked-end steel fibers (35 mm length, 0.55 mm diameter) to study the fracture and flexural behavior of SCC through compression, tension, and bending tests. Their results indicated an improvement in the serviceability limit state of SCC.

Moreover, traditional measurement methods such as strain gauges and extensometers are limited in their ability to capture the initiation, propagation, and coalescence of cracks, as they only provide macroscopic deformation data [23]. **Digital Image Correlation (DIC)**, a non-contact method for measuring full-field displacement and strain on material surfaces [24], has been increasingly used to analyze the dynamic crack propagation and strain evolution in concrete materials [25–28].

This study systematically investigates the mechanical properties of SCC reinforced with hybrid steel fibers of varying parameters using the DIC method. By designing experimental schemes with different fiber contents and hybrid ratios, combined with DIC technology, the effects of hybrid steel fibers on the compressive strength, flexural strength, and crack propagation behavior of SCC were analyzed. The results demonstrate that hybrid steel fibers effectively enhance the mechanical properties of SCC, with synergistic effects significantly improving toughness and crack resistance. The findings provide theoretical and experimental support for the design and application of hybrid steel fiber-reinforced SCC.

## 1. Materials and methods

### 1.1 Test materials and proportions

（1）Cement.The cement is P·O42.5 ordinary Portland cement produced by Chongqing Lafa base's dimensional cement formula. Its physical and mechanical properties are shown in **Table 1**.

（2）Fly ash. Fly ash is low calcium Class F fly ash produced and provided by Chongqing Hualuo Fly Ash Co., Ltd. Its physical and mechanical properties are shown in **Table 2**.

（3）Aggregate. The fine aggregate is limestone machine-made sand with a fineness modulus of 3.2, a fine powder content of 3%, an apparent density of 2712 kg/m³, and a bulk density of 1521 kg/m³, Coarse aggregate is 5~10mm crushed stone, with an apparent density of 2.78g/cm³ and a bulk density of 1.45g/cm³. Its physical and mechanical properties are shown in **Table 3** and **Table 4**.

（4）Steel fiber. The external form is as shown in **Fig 1**.The main performance is shown in **Table 5**.

（5）Water-reducing agent. Water reducing agent (WR) is a high-efficiency polycarboxylate water-reducing agent mother liquor with a solid content (mass fraction) of 50% to 60% and should be diluted 6.5 times.

（6）Water. Using tap water from the laboratory of Chongqing Jiaotong University.

**Table 1. Basic physical and mechanical properties of cement.**

| Apparent density/ (g·cm-3) | Specific surface area/ (m2·kg-1) | Sieve residual ratio(45μm sieve, by mass)/% | Water demand ratio for normal consistency (by mass)/% | Setting time/ min | | 3d strength/MPa | |
| --- | --- | --- | --- | --- | --- | --- | --- |
| | | | | Initial | Final | Compressive | Flexural |
| 3.1 | 345 | 8.4 | 26.80 | 165 | 226 | 30.0 | 5.90 |

**Table 2. Physical properties of fly ash.**

| Apparent density/ (g·cm-3) | Sieve residual ratio(45 μm sieve, by mass)/% | IL(by mass)/% | Water demand ratio(by mass)/% | Activity index/% | |
| --- | --- | --- | --- | --- | --- |
| | | | | 7d | 28d |
| 2300 | 0 | 5 | 95 | 83 | 95 |

**Table 3. Gradation of fine aggregate.**

| Sieve diameter/mm | Sieve residual ratio(by mass)/% | Cumulative sieve residual ratio(by mass)/% | Di/mm |
|---|---|---|---|
| 4.75 | 0 | 0 | 0 |
| 2.36 | 19.3 | 19.3 | 3.35 |
| 1.18 | 17.4 | 36.7 | 1.67 |
| 0.60 | 22.1 | 58.8 | 0.84 |
| 0.30 | 21.7 | 80.5 | 0.42 |
| 0.15 | 13.7 | 94.2 | 0.21 |
| 0.075 | 3.6 | 97.8 | 0.11 |
| 0 | 2.2 | 100 | 0.03 |

**Table 4. Gradation of coarse aggregate.**

| Sieve diameter/mm | Sieve residual ratio(by mass)/% | Cumulative sieve residual ratio(by mass)/% | Di/mm |
|---|---|---|---|
| 16 | 0 | 0 | 0 |
| 13.2 | 0.27 | 0.27 | 14.53 |
| 9.5 | 0.40 | 0.67 | 11.20 |
| 4.75 | 0.33 | 1.00 | 6.74 |

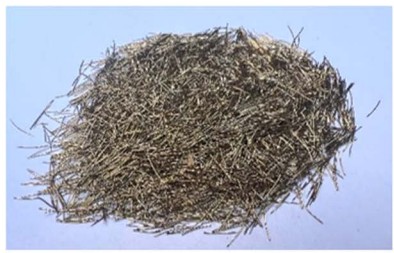 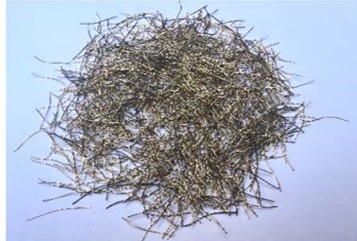 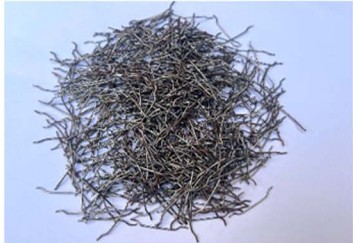

6mm 13mm 25mm

**Fig 1. Steel fibers with different aspect ratios.**

**Table 5. Types of steel fibers and main physical and mechanical properties.**

| Steel fiber types | Length/mm | Aspect ratio | Density (t/m³) | Elastic Modulus (GPa) | Tensile strength (MPa) |
|---|---|---|---|---|---|
| Ordinary steel fiber (hook-end) | 25 | 33 | 7.8 | 210 | 850 |
| Ordinary steel fiber | 13 | 65 | 7.8 | 240 | >850 |
| Ultra-short ultra-fine steel fiber | 6 | 35 | 7.8 | 240 | >850 |

（7）Mix ratio selection. The mix ratio of this test was carried out based on the mix ratio determined in the previous test study. **Table 6** records the mix ratio of steel fiber self-compacting concrete with different steel fiber volume content of 1m³.

(For the convenience of research, 3%6 represents 6mm steel fiber with a volume content of 3%, and other naming methods are the same)

Self-compacting steel fiber concrete should use a mandatory mixer. In this experiment, coarse aggregate, fine aggregate, and cement are first poured in, and then the steel fibers are slowly added in batches. The water mixed with the water-reducing agent is poured into the mixer. After the addition is completed, continue stirring for one minute. When there is a large amount of fiber or fiber, a fiber disperser can be used.

## 1.2 Experiment preparation

Before the test, the self-compacting concrete fluidity test and passing ability test were carried out [29]. The slump expansion diameter and J-ring expansion of self-compacting concrete with different volume dosages are shown in **Fig 2** and **Fig 3**.

The experimental data are shown in **Table 7**, according to the "Technical Specification for Application of Self-Compacting Concrete" JGJ/T 283–2012, the standard for the slump flow of self-compacting concrete is 500mm to 800mm.

The target value for the slump flow of self-compacting concrete should not be less than 500mm; therefore, the self-compacting concrete with various steel fibers meets the construction requirements.

## 1.3 Experimental method

In this study, the mechanical properties of self-compacting concrete were characterized by compressive strength, flexural strength, and elastic modulus, so cube specimens and beam specimens were prepared for corresponding

**Table 6. Mix ratio of steel fiber self-compacting concrete specimens (1m³/kg).**

| Number | Cement (kg) | Water (kg) | Fly ash (kg) | Coarse aggregate (kg) | Fine aggregate (kg) | Water reducing agent (%) | Steel fiber content (%) |
|---|---|---|---|---|---|---|---|
| 1 | 1 | 0.37 | 0.43 | 1.37 | 1.06 | 0.16% | 0 |
| 2 | 1 | 0.37 | 0.43 | 1.37 | 1.06 | 0.16% | 3%6mm |
| 3 | 1 | 0.37 | 0.43 | 1.37 | 1.06 | 0.16% | 3%13mm |
| 4 | 1 | 0.37 | 0.43 | 1.37 | 1.06 | 0.16% | 3%25mm |
| 5 | 1 | 0.37 | 0.43 | 1.37 | 1.06 | 0.16% | 1%6mm+1%13mm+1%25mm |

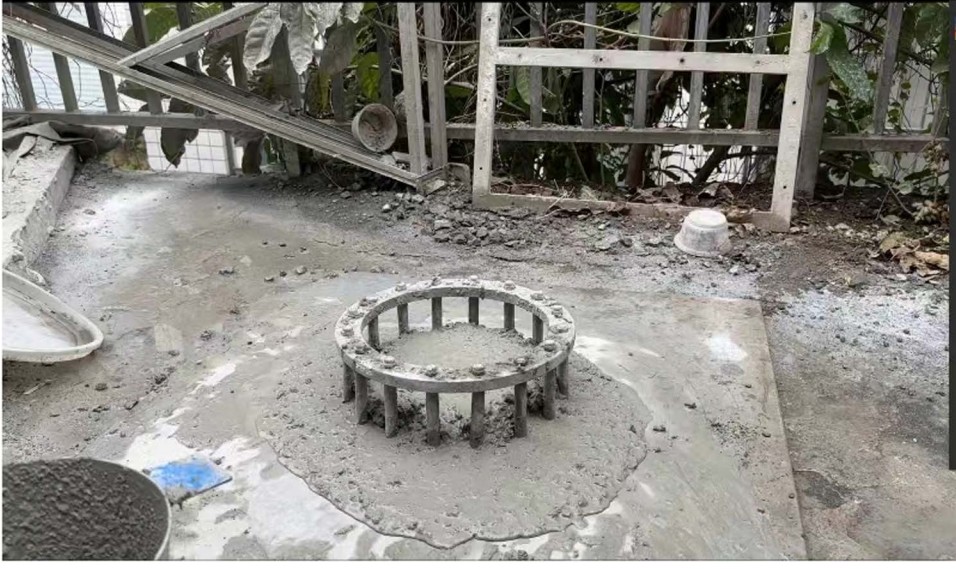

**Fig 2. Slump expansion of steel fiber self-compacting concrete.**

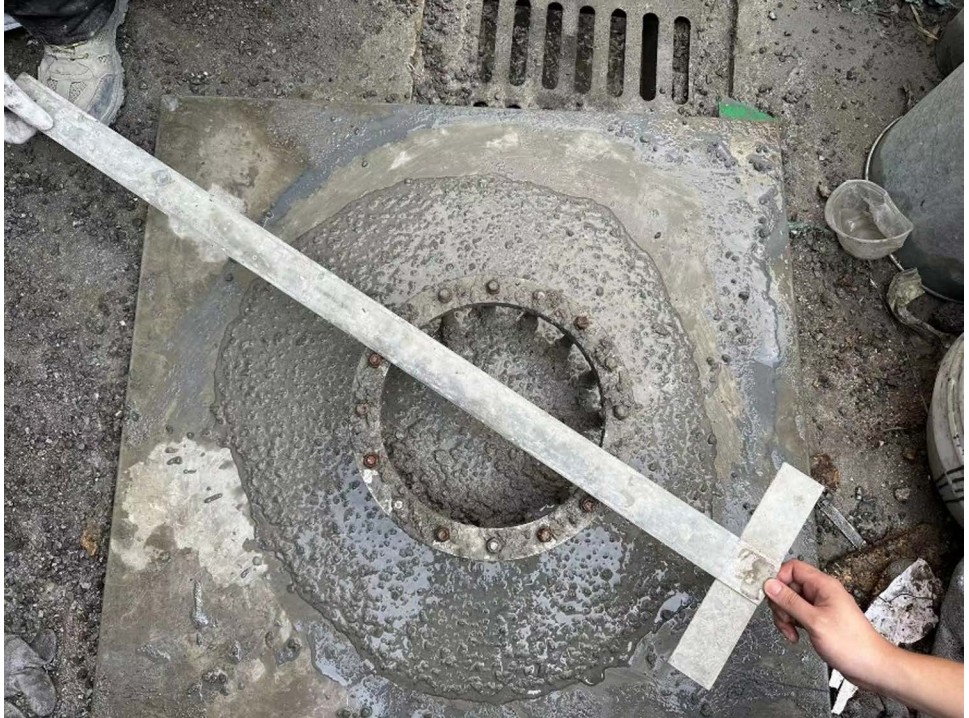

**Fig 3. J-ring expansion of steel fiber self-compacting concrete.**

**Table 7. Slump expansion diameter and J-ring expansion of steel fiber self-compacting concrete under different steel fiber content.**

| Number | Length | Fiber content | slump propagation (mm) | J ring expansion (mm) |
|---|---|---|---|---|
| S-0 | 0 | 0 | 840 | 800 |
| 3%6 | 6mm | 3%6mm | 660 | 500 |
| 3%13 | 13mm | 3%13mm | 520 | 410 |
| 3%25 | 25mm | 3%25mm | 730 | 610 |
| 6+13+25 | 6mm+13mm+25mm | 1%6mm+1%13mm+1%25mm | 650 | 530 |

tests. This experiment conducted a preliminary experimental study on five groups of specimens: self-compacting concrete, single-content steel fiber concrete, and self-compacting concrete mixed with ordinary steel fiber and hook-end steel fiber at 3% content. Comparative analysis of the effect of steel fibers on the properties of self-compacting concrete. The number of test specimens for each group of solutions is 30 cube compressive strength test, flexural test, and bending and tensile test. The total number of specimens is 90. The size and quantity of specimens are summarized in **Table 8**.

The loading equipment used in the experiment is an MTS hydraulic universal testing machine. Different loading rates were set for different tests, and load and displacement data were collected during the loading process. The strain monitoring equipment is a non-contact 3D full-field strain measurement and analysis system, XTDIC. Industrial cameras were used to capture speckle image information on the surface of the test specimens. Through the acquired speckle image information, DIC analysis was performed to obtain full-field displacement and strain data during the loading process.

**Table 8. Dimensions and quantity of specimens.**

| Test content | Specimen size (mm) | Number of test pieces | Number of samples |
|---|---|---|---|
| Compressive strength | 100×100×100 | 6 | 30 |
| Flexural strength | 100×100×400 | 6 | 30 |
| Elastic Modulus | 100×100×400 | 6 | 30 |

## 2．Results and analysis

### 2.1 Cube compressive strength

**Fig 4** shows the compression failure modes of ordinary concrete, single dosage, and hybrid steel fiber concrete cubes.

According to the compression failure pattern of the concrete specimen in **Fig 4**, it can be seen that when the self-compacting concrete specimen is mixed with steel fibers, its peeling speed slows down under compressive load, the time it continues to bear pressure after peeling increases, and the peeling is more uniform.

According to the test, the data shown in **Table 9** is obtained. The data in the table shows that the addition of steel fibers can effectively improve the compressive strength of self-compacting concrete. Compared with ordinary self-compacting

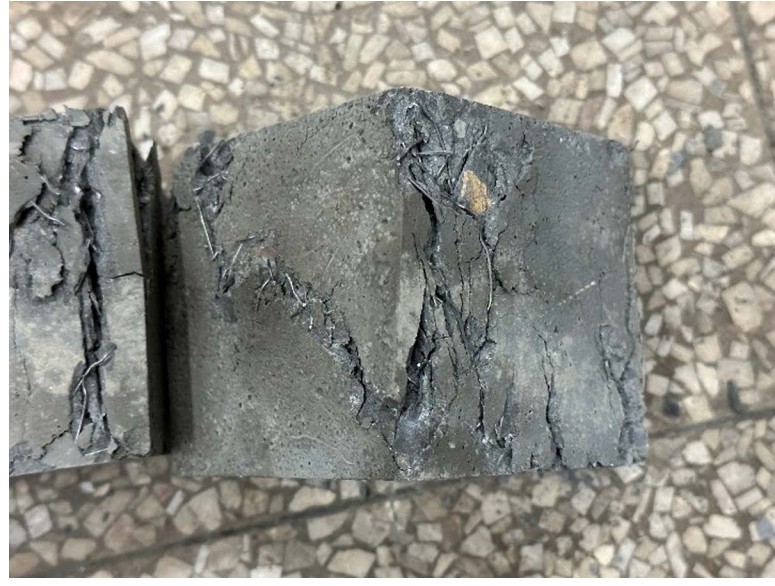

**Fig 4. Compression failure pattern of concrete cube specimens.**

**Table 9. Cube compressive strength test results (28d, MPa).**

| Number | Fiber content | Compressive strength (MPa) | Improvement |
|---|---|---|---|
| S-0 | 0 | 73.65 | — |
| 3%6 | 3%6mm | 82.56 | 12.09% |
| 3%13 | 3%13mm | 85.28 | 15.79% |
| 3%25 | 3%25mm | 83.26 | 13.05% |
| 6+13+25 | 1%6mm+1%13mm+1%25mm | 103.96 | 41.15% |

concrete, when the fiber content is 3%, the increase in the compressive strength of concrete with different aspect ratios is not much different, Among them, the addition of 13mm steel fiber greatly improves the compressive strength of self-compacting concrete. By mixing three types of steel fibers into self-compacting concrete, the compressive strength of the concrete has been significantly improved. Compared with ordinary self-compacting concrete, the compressive strength has increased by 41.15%. As shown in **Fig 5**, when the volume fraction of steel fibers is the same, the compressive strength of hybrid steel fiber concrete is increased by about 20% compared with steel fiber alone. But overall, the addition of steel fibers has a limited effect on enhancing the compressive strength of self-compacting concrete.

## 2.2 Flexural strength

**Fig 6** shows the flexural failure morphology of ordinary concrete, single dosage, and hybrid steel fiber concrete cubes.

It can be seen from the failure shape that under the ultimate load, cracks soon appeared at the edge of the tensile zone at the bottom of the concrete specimen without adding steel fibers. Under the influence of stress concentration and tip effect, the crack immediately expanded to the top, and a clear crack appeared. At the same time, several concrete blocks fell off in the tension area, accompanied by a crisp and loud breaking sound, plain concrete blocks quickly failed and stopped working. The flexural concrete test block with steel fiber added will have a slower time of cracking than the plain concrete test block. Under the action of concentrated force, a main crack first appears in the bottom tension zone. As the load increases, the main crack gradually expands from the bottom of the tension zone to the top of the beam until the specimen is completely cracked. The flexural strength obtained through the test is shown in **Table 10**.

As can be seen from **Fig 7**, steel fibers can effectively improve the flexural strength of self-compacting concrete. Compared with ordinary self-compacting concrete, the flexural strength of self-compacting concrete mixed with fibers has a greater increase. When the volume content is 3%, the 13mm steel fiber has the most obvious effect on enhancing the flexural strength of self-compacting concrete, while the addition of ultra-short and ultra-fine 6mm steel fibers has no significant

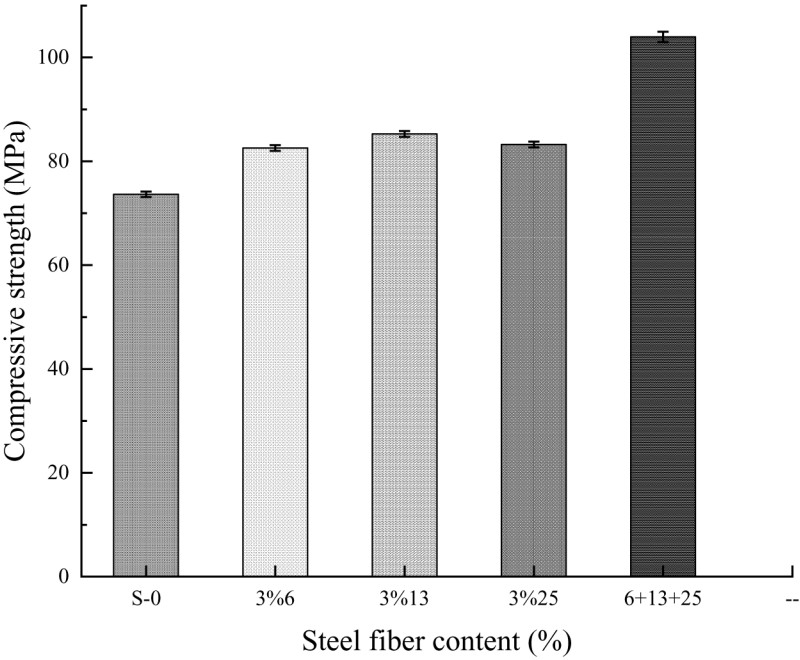

**Fig 5. The relationship between the compressive strength value and the volumetric content of steel fiber.**

effect on the flexural strength of self-compacting concrete. This shows that as the aspect ratio of steel fibers increases, the flexural strength of self-compacting concrete continues to increase. After mixing three steel fibers with different aspect ratios, mixing 1%6mm+1%13mm+1%25mm into self-compacting concrete, it can be seen that the reinforcing effect of mixed fibers on self-compacting concrete is not as good as the reinforcing effect of 13mm steel fiber alone. Analyze the reasons: Steel fibers with different aspect ratios within the concrete are not easily dispersed fully and evenly, and the agglomeration of the fibers with cement slurry or aggregate increases the internal pores of the specimen, resulting in a limitation in the enhancement effect of mixed fibers on the flexural strength of self-compacting concrete.

## 2.3 Elastic modulus

According to the test results, it can be seen that ordinary self-compacting concrete specimens have very long cracks after being loaded. Running through the entire rectangular parallelepiped, the crack develops from top to bottom from the upper platform, and the width of the crack is very wide when it is close to the upper platform, and is narrower near the lower platform. The cracks in the mixed steel fiber specimens have a narrower width, develop slower, and can withstand the load for a longer time. The failure modes are shown in **Fig 8**.The elastic modulus obtained through the test is shown in **Table 11**.

It can be seen from **Fig 9** that when steel fibers are added, the elastic modulus of self-compacting concrete is significantly improved. At the same dosage, steel fibers with lengths of 6mm and 13mm improve the elastic modulus of self-compacting concrete more than the addition of 25mm steel fibers, indicating that the performance improvement of self-compacting concrete by hook-end steel fibers is weak. It can be seen from the **Fig 9** that hybrid steel fibers have a

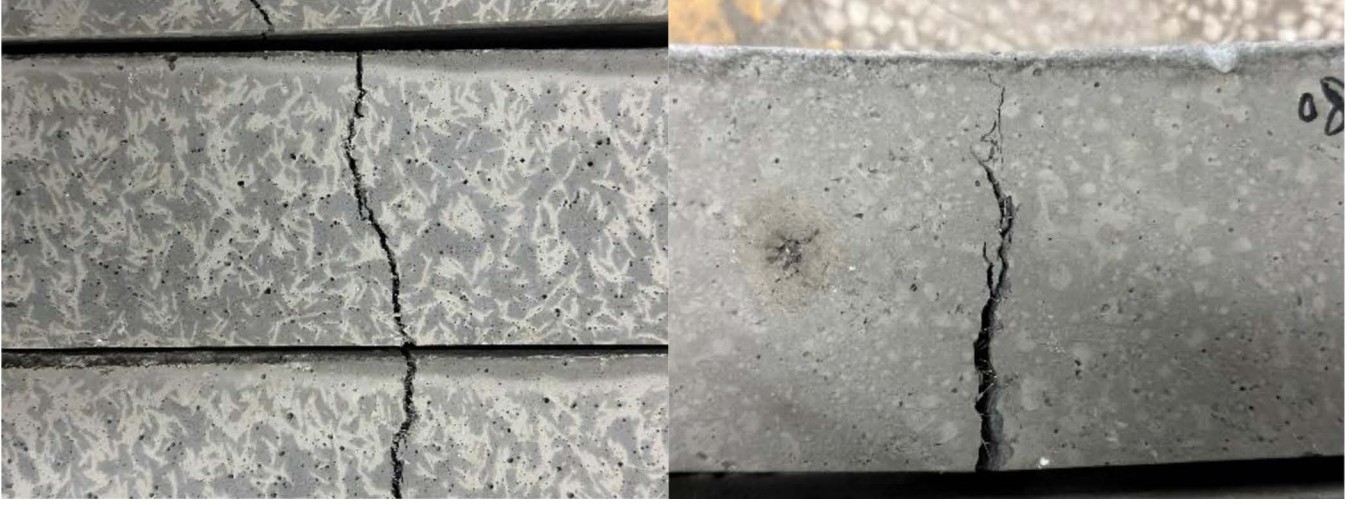

**Fig 6. Concrete flexural failure pattern.**

**Table 10. Flexural strength test results (28d, MPa).**

| Number | Fiber content | Flexural strength (MPa) | Improvement |
|---|---|---|---|
| S-0 | 0 | 3.93 | — |
| 3%6 | 3%6mm | 8.41 | 113.95% |
| 3%13 | 3%13mm | 14.96 | 280.66% |
| 3%25 | 3%25mm | 12.24 | 211.45% |
| 6+13+25 | 1%6mm+1%13mm+1%25mm | 14.47 | 268.19% |

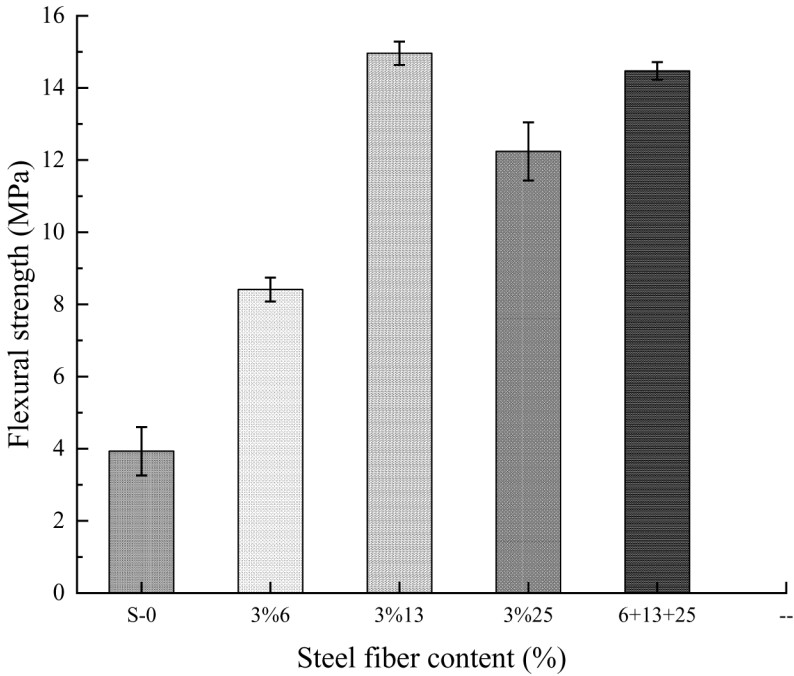

**Fig 7. The relationship between the tensile elastic modulus and the volumetric content of steel fiber.**

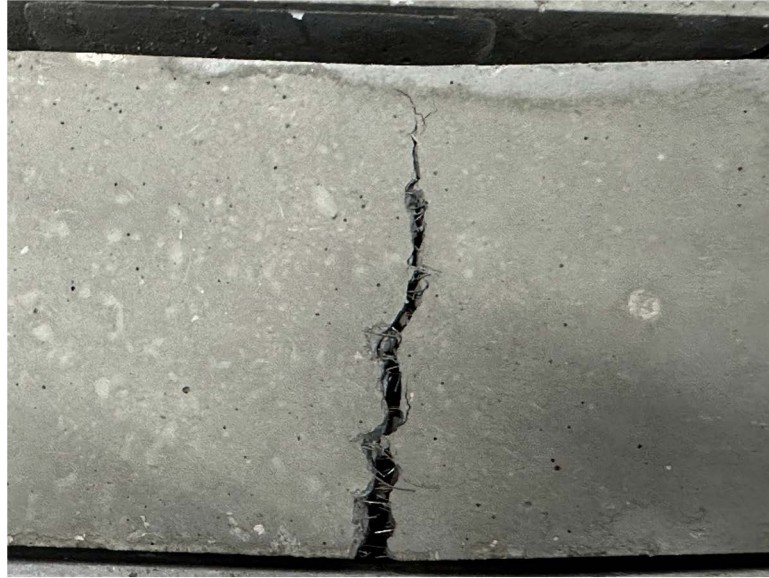

**Fig 8. Elastic Modulus test loading of self-compacting steel fiber concrete.**

more obvious increasing trend in the elastic modulus of self-compacting concrete. This shows that the incorporation of hybrid steel fibers has a certain effect on improving the elastic modulus, but generally speaking, steel fibers have little effect on the elastic modulus.

## 2.4 Tension-compression ratio

To study the impact of steel fiber on the strength of concrete, this article further analyzed the test data conducted previously to obtain the tension-to-compression ratio of the specimen and plotted it with the steel fiber content as the X-axis and the tension-to-compression ratio as the Y-axis. As shown in **Fig 10**, The Y-axis data is the ratio of the flexural strength of the specimen to the compressive strength of the specimen.

The influences of steel fibers with different lengths and contents on the tension-compression ratio of concrete are diverse. Among them, the 13mm steel fibers with a volume content of 3% can enhance the tension-compression ratio of

**Table 11. Elastic modulus test results (28d, MPa).**

| Number | Initial load $F_0 (kN)$ | Test piece pressure area $A(mm^2)$ | Average elastic modulus $E_c(MPa)$ |
|---|---|---|---|
| S-0 | 10.0 | 10000 | 26490 |
| 3%6 | 10.0 | 10000 | 34880 |
| 3%13 | 10.0 | 10000 | 35230 |
| 3%25 | 10.0 | 10000 | 32820 |
| 6+13+25 | 10.0 | 10000 | 40820 |

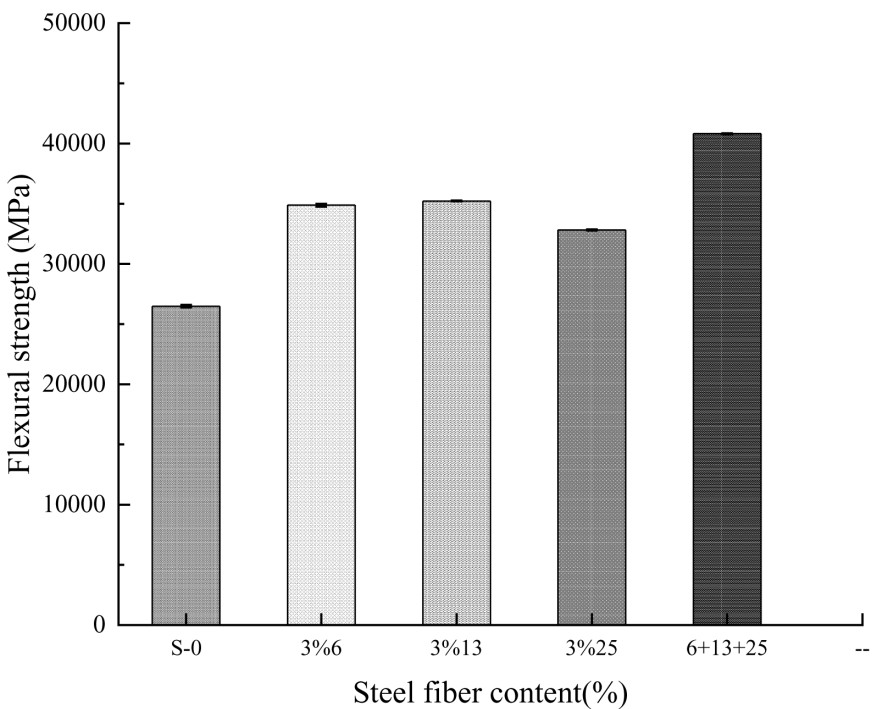

**Fig 9. The relationship between the elastic modulus and the volumetric content of steel fiber.**

concrete to the greatest extent. While the effect of mixing steel fibers of multiple lengths is relatively weaker, it is still better than the situation without steel fibers and some cases of single-length steel fibers. Consequently, steel fibers can effectively improve the tension-compression ratio of self-compacting concrete.

## 2.5 Analysis of the evolution process of surface strain fields

Taking some specimens as examples, the evolution characteristics of the surface strain field under different load levels are analyzed. The **Fig 11** illustrates the evolution process of the surface strain field in SFRSCC specimens under uniaxial compression as the loading stress level increases. From the figure, it can be observed that during the initial stage of the test, the specimen surface remains intact, and the specimen is in the elastic deformation stage, with the maximum principal strain field uniformly distributed. As the load increases, flocculent strain concentration zones appear on the specimen surface. With further loading, the strain concentration on the surface of the HFR-RCAC specimen becomes increasingly pronounced, accompanied by the development of macroscopic cracks and partial surface spalling. Subsequently, macroscopic cracks appear in most regions of the specimen, leading to unstable failure and exhibiting clear brittle failure characteristics.

Observations reveal that the macroscopic crack propagation process in SFRSCC specimens is similar. Before the formation of macroscopic failure surfaces, strain concentration phenomena occur in multiple regions of the specimens. Microcracks initially initiate and propagate in the corner regions of the specimens, then extend along the diagonal path toward the upper and lower surfaces until they penetrate the entire surface of the specimen. By comparing the full-field strain images of specimens with 3% 6mm, 3% 13mm, 3% 25mm, and the hybrid 6+13+25mm fibers, it is evident that specimens with single-type steel fibers develop microcracks in most regions. The crack propagation paths in these specimens are more complex compared to the single-crack path in the hybrid fiber specimen, and the macroscopic failure is more severe. This indicates that the incorporation of steel fibers improves the strain distribution in concrete, making the strain distribution under compression more uniform and thereby altering the failure mode of the specimens.

Notably, after the addition of steel fibers, the macroscopic crack propagation stage in the specimens is relatively prolonged. This is because steel fibers can bear part of the tensile stress after concrete cracking, effectively inhibiting the initiation and propagation of microcracks within the SFRSCC specimens. The maximum principal strain on the surface of the

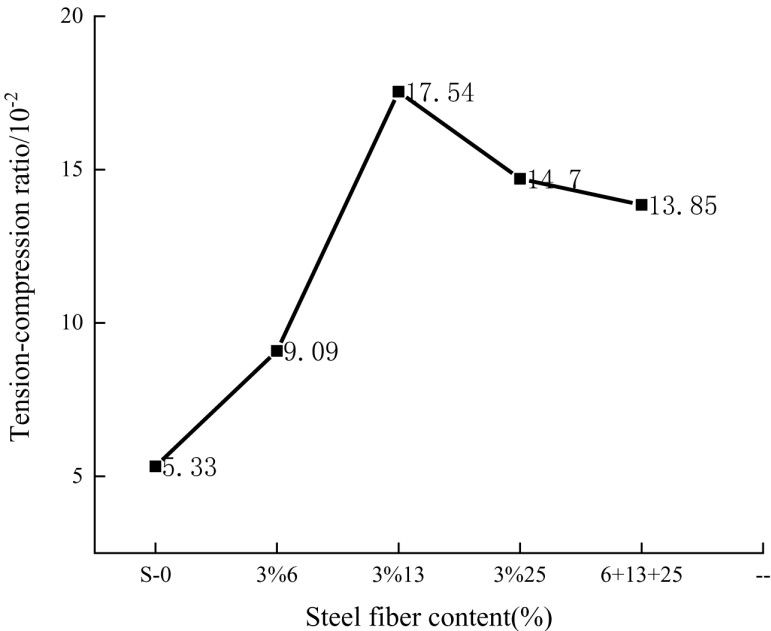

**Fig 10. The relationship between the tension-compression ratio and the volume content of steel fiber.**

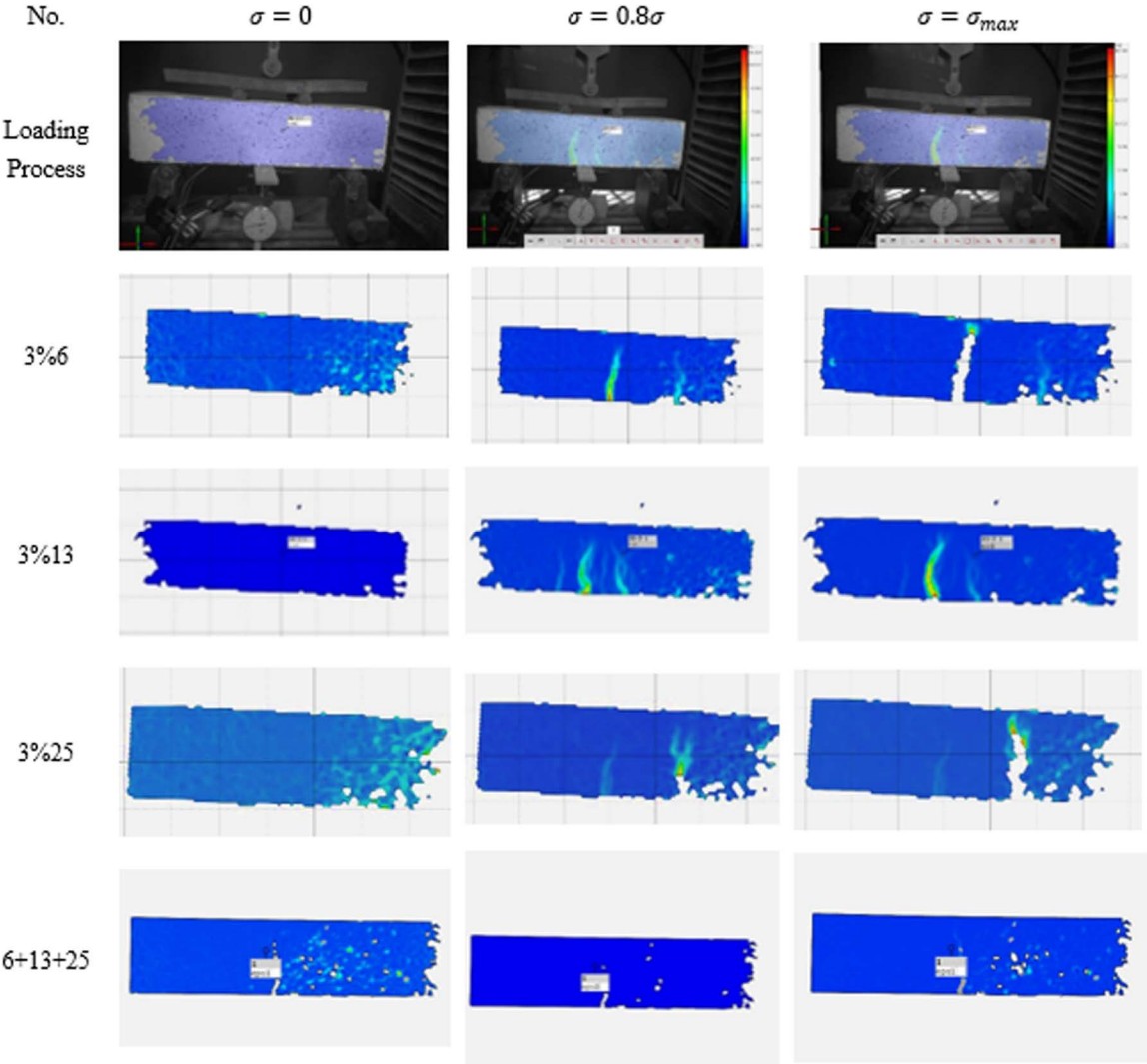

**Fig 11. Evolution process of surface strain field at different stages of cubic compressive test.**

SFRSCC specimens exhibits a linear distribution, and as the load continues to increase, this "line" rapidly expands, eventually forming a failure surface. From the perspective of energy dissipation [30], due to the incorporation of steel fibers, the energy consumed by microcrack propagation within the specimens includes not only the energy consumed by the failure of the interfacial transition zone in the concrete matrix but also a significant amount of energy consumed during the tensile deformation and eventual failure (fracture or pull-out) of the steel fibers. Therefore, the enhancement effect of steel fibers on concrete is primarily manifested in hindering the propagation of macroscopic cracks, thereby reinforcing the matrix.

## 3 . Discussion

### 3.1 Optimal dosage

According to the compression, flexural, and elastic modulus test results, it can be seen that the addition of steel fibers can effectively improve the mechanical properties of self-compacting concrete. Since the flexural strength of 13mm steel fiber

alone increases significantly, it is difficult to determine the specific mixing ratio that will have the most obvious improvement effect on self-compacting concrete.

In order to find the optimal blending ratio, data normalization is used to narrow the selection criteria to between 0 and 1. By accumulating the regularization criterion values, the comprehensive impact of steel fibers on self-compacting concrete is determined.

The data normalization formula is as follows,

$$D_n = \frac{D - D_{min}}{D_{max} - D_{min}}$$

(3-1)

In the formula:

$D$--- the original value of the test data,

$D_{max}$ --- the maximum value of the test data,

$D_{min}$ --- the minimum value of the test data,

$D_n$ ---the normalized value of the test data.

In this study, the experimental data obtained above were imported into the Origin software, and the comprehensive impact of steel fibers on self-compacting concrete was obtained through normalized calculations, as shown in **Fig 12**. It can be seen from the normalized data that when the fiber content is 1%6mm+1%13mm+1%25mm, the improvement effect on the mechanical properties of self-compacting concrete is the best. It can be seen that hybrid steel fibers can effectively inhibit the development of concrete cracks and improve the cracking resistance of self-compacting concrete.

### 3.2 Comparison

To prove the reliability of this study, this article selects three similar studies from previous years for comparative analysis. In 2019, researchers such as Alabduljabbar [31] added steel fibers with volume fractions of 0, 1%, 1.5%, and 2% to self-compacting concrete respectively. Research results show that when the volume content is 2%, the compressive strength and tensile strength of steel fiber reinforced self-compacting concrete reach the peak. Ghorbani (2020)[32] added steel fibers with a volumetric content of 0~1.65% to self-compacting concrete for research. The results showed that the performance of self-compacting concrete is optimal when the content is 1.65%. Abbas (2021)[33] also added steel fibers with a volume content of 0~2% in self-compacting concrete, the test results show that when the dosage is 2%, the self-compacting concrete has the highest strength. The specific information is as shown in **Table 12**. Based on the above three test results, this study comparatively analyzes the reinforcing effects of steel fibers at different dosages.

It can be seen from the growth data listed in Table 13 that different steel fiber content has different reinforcing effects on self-compacting concrete. Compared with other experimental research results, the mechanical properties of steel fiber reinforced self-compacting concrete obtained in this study have obvious hybrid effects. This study explains the reinforcement mechanism of self-compacting concrete by hybrid steel fibers with different aspect ratios. It can be seen from the data in the table that this research shows strong performance advantages in both compressive strength and flexural strength. Higher compressive strength means that the material is less likely to be damaged when subjected to pressure and can be applied to scenarios with higher requirements for bearing capacity. The extremely high flexural strength indicates that the material has better toughness and stability when subjected to bending forces, and is suitable for structures or components that need to resist bending deformation. This makes the research results potentially have wider applicability and competitiveness in practical applications.

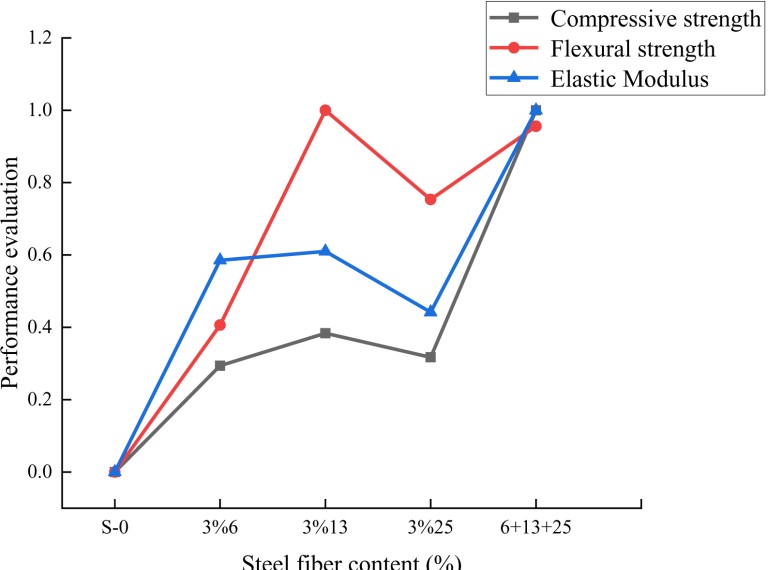

**Fig 12. Comprehensive effect of fiber content on self-compacting concrete.**

**Table 12. Comparison of experimental data.**

| Refs. | Steel fiber percentages(%) | Days | Opti-mum(%) | Compression strength (MPa) | Flexure strength (MPa) |
|---|---|---|---|---|---|
| Alabduljabbar et al.,(2019) | 0~2 | 28 | 2 | +17.2% | +1.96% |
| Ghorbani et al.,(2020) | 0~1.65 | 28 | 1.65 | -1.51% | +28.57% |
| Abbas et al.,(2021) | 0~2 | 28 | 2 | +40.0% | – |
| This Research | 0~3 | 28 | 3 | +41.15% | 268.19% |

## 4. Conclusion

（1）When mixed steel fibers with a volume content of 1% 6mm + 1% 13mm + 1% 25mm are added to the concrete, the compressive strength of self-compacting concrete increases by 41.15%; the flexural strength increases by 268.19% respectively. Therefore, the compressive and flexural strength of self-compacting concrete can be improved by incorporating hybrid steel fibers.

（2）Change in Failure Mode. Based on the evolution process of the maximum principal strain, the dynamic evolution characteristics of macro- and micro-cracks on the surface of SFRSCC specimens were analyzed. Before the formation of macroscopic cracks, strain concentration phenomena occur in multiple regions of the specimens. The macroscopic crack propagation stage is relatively prolonged, and the crack paths are more complex. In contrast, plain concrete specimens exhibit a single microcrack propagation path. Microcracks initially initiate and propagate in the diagonal regions of the specimens, then rapidly extend along the diagonal direction, leading to brittle failure. The macroscopic crack propagation stage is very short.

（3）When steel fibers are added to the concrete, the test block shows obvious ductility during the loading process, which is manifested as continuous cracking of the test piece.

It can be seen that the basic mechanical properties of the self-compacting hybrid steel fiber concrete obtained by mixing three different aspect ratio steel fibers meet the requirements of self-compacting concrete as the fiber volume content increases. It not only achieves self-compacting properties that are difficult to achieve at high dosages, but also meets the workability of the mixture and complies with national standards. Secondly, after multiple experimental tests, the self-compacting hybrid steel fiber concrete test block has good compressive and flexural properties. Moreover, steel fibers can significantly inhibit the development of macro cracks, making concrete more resistant to cracking.

## Author contributions

**Conceptualization:** Hang Hu, Jianyi Gu, Fuwei Zhu.

**Data curation:** Hang Hu.

**Formal analysis:** Hang Hu.

**Funding acquisition:** Fuwei Zhu.

**Investigation:** Yang Tu.

**Methodology:** Yang Tu.

**Project administration:** Yang Tu.

**Resources:** Jianyi Gu.

**Software:** Yi Zhou.

**Supervision:** Yi Zhou.

**Validation:** Hang Hu, Jianyi Gu.

**Visualization:** Fuwei Zhu.

**Writing – original draft:** Hang Hu.

**Writing – review & editing:** Jianyi Gu.

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
