## [Decision Letter · Decision Letter 0]

29 Dec 2024

PONE-D-24-55030The Influence of Hybrid Fiber on the Mechanical Properties of Self-Compacting ConcretePLOS ONE

Dear Dr. Gu,

Thank you for submitting your manuscript to PLOS ONE. After careful consideration, we feel that it has merit but does not fully meet PLOS ONE’s publication criteria as it currently stands. Therefore, we invite you to submit a revised version of the manuscript that addresses the points raised during the review process.

**ACADEMIC EDITOR:** Based on the Reviewers' comments, it is suggested that a MAJOR revision be provided. It is obligatory that rebutals to all commnets and suggestions be adopted and solved. Additionaly, it is hardly suggested that reproducibility (duplicate and triplicate) must be provided.

We look forward to receiving your revised manuscript.

Kind regards,

Wislei Riuper Osório

Academic Editor

PLOS ONE

Journal Requirements:

Please confirm at this time whether or not your submission contains all raw data required to replicate the results of your study. Authors must share the “minimal data set” for their submission. PLOS defines the minimal data set to consist of the data required to replicate all study findings reported in the article, as well as related metadata and methods (https://journals.plos.org/plosone/s/data-availability#loc-minimal-data-set-definition ).

If your submission does not contain these data, please either upload them as Supporting Information files or deposit them to a stable, public repository and provide us with the relevant URLs, DOIs, or accession numbers. For a list of recommended repositories, please see https://journals.plos.org/plosone/s/recommended-repositories .

Additional Editor Comments:

Based on the Reviewers' comments, it is suggested that a MAJOR revision be provided. It is obligatory that rebutals to all commnets and suggestions be adopted and solved. Additionaly, it is hardly suggested that reproducibility (duplicate and triplicate) must be provided.

Reviewers' comments:

Reviewer's Responses to Questions

**Comments to the Author**

1. Is the manuscript technically sound, and do the data support the conclusions?

Reviewer #1: Partly

Reviewer #2: Partly

2. Has the statistical analysis been performed appropriately and rigorously? 

Reviewer #1: No

Reviewer #2: No

3. Have the authors made all data underlying the findings in their manuscript fully available?

Reviewer #1: No

Reviewer #2: No

4. Is the manuscript presented in an intelligible fashion and written in standard English?

Reviewer #1: No

Reviewer #2: No

5. Review Comments to the Author

Reviewer #1: Complete review of English – adjust verb tenses.

Correct the font numbering in the text, placing it in ascending order.

The introduction must include the most current references, preferably from the last 5 years.

Cite the standards for all tests performed in this study.

Table 4 Gradation of coarse aggregate - to correct

Table 7 – cite the limits of self-compacting concrete standards for each measured property and classify your concretes

Tables 9, 10, 11 – insert standard deviation, variance

Line 172 – “and shut down the fuel tank” - I don’t understand?

Correct equation 1.2

Regarding the Elastic modulus test: Specify the size of the test piece and the standard used to perform the test. Show a photo of the execution of this test.

Include in the article a statistical analysis of the results.

Reviewer #2: Make a careful review of the (tenses) verbs and verb agreements in the entire article.

Correct the numbering of bibliographic references cited in the Article in ascending order – [1], [2], [3], etc.

Mention all standards used in all tests carried out in this research.

Table 7 – mention the limits of self-compacting concrete standards for each measured property and classify the traits of your concrete.

Tables 9, 10, 11 and 12 – and corresponding Figures 8, 9, 10 and 11 – insert all values of standard deviation, variance.

Correct equation (1.2) and the line symbol line 185 – ff

Cite the reference and norm of equation (1-4).

281 2.4 Tension-compression ratio

Present three current bibliographical references (last five years) that corroborate this methodology to evaluate the impacts of steel fiber or steel fibers on the strength of concrete results.

Cite the reference and norm of equation (3-1).

Prepare a statistical analysis of the results obtained.

6. PLOS authors have the option to publish the peer review history of their article (what does this mean? ). If published, this will include your full peer review and any attached files.

**Do you want your identity to be public for this peer review?** For information about this choice, including consent withdrawal, please see our Privacy Policy .

Reviewer #1: No

Reviewer #2: No

---

## [Author Response · Author response to Decision Letter 1]

31 Jan 2025

Thank you for pointing this out. I agree with these comments. The tenses in the whole article have been carefully examined. The reference numbers cited in the article have been corrected and arranged in ascending order. The standards used in all the tests in the article have been clearly marked. Regarding Table 7, the limits of each measurement attribute in the self-compacting concrete standard have been mentioned in the article. Since there is only one set of performance parameters under different mixtures in the article, the calculation of standard deviation and variance is not possible. The mistakes within the formulas have been fixed, the bibliographic information has been renewed, and the outcomes of the article have been distinctly stated.

---

## [Editor Report · Decision Letter 1]

2 Feb 2025

PONE-D-24-55030R1The Influence of Hybrid Fiber on the Mechanical Properties of Self-Compacting ConcretePLOS ONE

Dear Dr. Gu,

Thank you for submitting your manuscript to PLOS ONE. After careful consideration, we feel that it has merit but does not fully meet PLOS ONE’s publication criteria as it currently stands. Therefore, we invite you to submit a revised version of the manuscript that addresses the points raised during the review process.

We look forward to receiving your revised manuscript.

Kind regards,

Wislei Riuper Osório

Academic Editor

PLOS ONE

Journal Requirements:

Additional Editor Comments:

Dear Authors;

It can be observed that majority of questions /comments provided by Reviewers are solved and improved. However, in my frank opinion, the manuscript needs other improvements before its final publication.

Firstly, all Tables need obligatory be revised and reworked. This in order to included error ranges. Secondly, all plots and graphs should obligatory be reworked to included error bars. For this purposes minus and maximum bars should be depicted.

Additionally, all imagens concerning to Experiemntal procedure section should be revised and scale bars obligatory be also included.

---

## [Author Response · Author response to Decision Letter 2]

8 Feb 2025

The authors would like to thank the reviewers for their suggestions. All tables have been revised, and all figures have been redesigned, with error bars or numerical values added to the corresponding data as indicated.

---

## [Decision Letter · Decision Letter 2]

19 Feb 2025

PONE-D-24-55030R2The Influence of Hybrid Fiber on the Mechanical Properties of Self-Compacting ConcretePLOS ONE

Dear Dr. Gu,

Thank you for submitting your manuscript to PLOS ONE. After careful consideration, we feel that it has merit but does not fully meet PLOS ONE’s publication criteria as it currently stands. Therefore, we invite you to submit a revised version of the manuscript that addresses the points raised during the review process.

**Based on the first and second rounds of revisions is evidenced that a MAJOR REVISION be provided. It remarkable that  other two different Reviewers have also recommended a MAJOR revision. Also, it is strongly recommended that these two Reviewers be taken in account. Otherwise, no new round is indicated if these rebuttals are not provide. Please, consider fully other MAJOR considerations.**

We look forward to receiving your revised manuscript.

Kind regards,

Wislei Riuper Osório

Academic Editor

PLOS ONE

**Additional Editor Comments:**

Based on the first and second rounds of revisions is evidenced that a MAJOR REVISION be provided. It remarkable that other two different Reviewers have also recommended a MAJOR revision. Also, it is strongly recommended that these two Reviewers be taken in account. Otherwise, no new round is indicated if these rebuttals are not provide. Please, consider fully other MAJOR considerations.

Reviewers' comments:

Reviewer's Responses to Questions

**Comments to the Author**

1. If the authors have adequately addressed your comments raised in a previous round of review and you feel that this manuscript is now acceptable for publication, you may indicate that here to bypass the “Comments to the Author” section, enter your conflict of interest statement in the “Confidential to Editor” section, and submit your "Accept" recommendation.

Reviewer #3: (No Response)

2. Is the manuscript technically sound, and do the data support the conclusions?

Reviewer #3: Partly

3. Has the statistical analysis been performed appropriately and rigorously? 

Reviewer #3: No

4. Have the authors made all data underlying the findings in their manuscript fully available?

Reviewer #3: No

5. Is the manuscript presented in an intelligible fashion and written in standard English?

Reviewer #3: No

6. Review Comments to the Author

**Reviewer #3: ** There are other studies with approaches/focuses very similar to this manuscript. Therefore, the authors need to highlight the difference in this study... what the scientific advance is.

The introduction needs more information, with more specific data and greater depth, since what is presented is already known information... it does not bring anything new.

The authors need to highlight the importance of this manuscript, presenting other data such as a schematic figure or numerical/visual simulation that represents the macrostructure of the concrete with the distribution of the steel fiber blend, justifying the increase in mechanical properties.

In addition, present graphs with test data, for example, stress x deformation, which is very important to understand the behavior of concrete with fibers, in this case, self-compacting.

The writing in English needs to be improved... in addition, there are many loose sentences (without connection) in the text.

The authors must perform a statistical analysis to assess whether the results of the mechanical properties of the concretes are statistically equal or not, considering the contents and types of steel fibers added.

7. PLOS authors have the option to publish the peer review history of their article (what does this mean? ). If published, this will include your full peer review and any attached files.

**Do you want your identity to be public for this peer review?** For information about this choice, including consent withdrawal, please see our Privacy Policy .

Reviewer #3: No

---

## [Author Response · Author response to Decision Letter 3]

20 Mar 2025

Dear Reviewer,

Thank you very much for your valuable comments and suggestions, which have helped us significantly improve the quality of our manuscript. Below, we provide a point-by-point response to your comments:

Comment 1: Highlight the difference in this study and its scientific advance.

Response:

We sincerely appreciate your suggestion. In the revised manuscript, we have emphasized the unique aspects of our study in the Introduction and Discussion sections. Specifically, our research focuses on the synergistic effects of hybrid steel fibers (1%6mm + 1%13mm + 1%25mm) on the mechanical properties of self-compacting concrete, which has not been systematically investigated in previous studies. Additionally, we employed the Digital Image Correlation (DIC) method to capture the full-field strain evolution and crack propagation, providing new insights into the macro- and micro-crack dynamics of hybrid fiber-reinforced concrete.

Comment 2: The introduction needs more information with specific data and greater depth.

Response:

We have revised the Introduction section to include more specific data and a deeper discussion of the current state of research. We have added recent references on steel fiber-reinforced concrete and highlighted the limitations of existing studies, such as the lack of systematic investigation into hybrid steel fibers.

Comment 3: Highlight the importance of the manuscript with schematic figures or numerical simulations.

Response:

Thank you for this suggestion. In the revised manuscript, the full-field strain evolution during specimen failure was obtained using the DIC method, and the dynamic evolution characteristics of macro- and micro-cracks on the specimen surface were analyzed, demonstrating the mechanical behavior of concrete under different fiber contents.

Comment 4: Improve the English writing and logical flow.

Response:

We have carefully revised the manuscript to improve the English language and ensure logical coherence.

Comment 5: Perform statistical analysis to assess the results.

Response:

We have conducted a statistical analysis to evaluate the significance of the differences in mechanical properties among the tested concrete samples. The results of this analysis are presented in the revised manuscript, and we have discussed their implications in the Results and Discussion sections.

Once again, we thank you for your constructive feedback, which has greatly improved the quality of our work. We hope that the revised manuscript meets your expectations.

Sincerely,

Jianyi Gu

---

## [Editor Report · Decision Letter 3]

24 Mar 2025

Research on the Mechanical Properties of Hybrid Steel Fiber-Reinforced Self-Compacting Concrete Based on the DIC Method

PONE-D-24-55030R3

Dear Dr. Gu,

We’re pleased to inform you that your manuscript has been judged scientifically suitable for publication and will be formally accepted for publication once it meets all outstanding technical requirements.

Kind regards,

Wislei Riuper Osório

Academic Editor

PLOS ONE
---

## [Editor Report · Acceptance letter]

PONE-D-24-55030R3

PLOS ONE

Dear Dr. Gu,

I'm pleased to inform you that your manuscript has been deemed suitable for publication in PLOS ONE. Congratulations! Your manuscript is now being handed over to our production team.

Kind regards,

on behalf of

Dr. Wislei Riuper Osório

Academic Editor

PLOS ONE